# Antibody escape, the risk of serotype formation, and rapid immune waning: Modeling the implications of SARS-CoV-2 immune evasion

**Catherine Albright[1], Debra Van Egeren[2], Aditya Thakur[1], Arijit Chakravarty[3], Laura F. White[4], Madison Stoddard[3] ***

1 Boston University, Boston, MA, United States of America, 2 Stanford University School of Medicine, Stanford, CA, United States of America, 3 Fractal Therapeutics, Cambridge, MA, United States of America, 4 Boston University School of Public Health, Boston, MA, United States of America

* madison.stoddard@fractaltx.com

**Data Availability Statement:** Model equations are contained within the manuscript. Code used to generate the figures (with the exception of

## Abstract

As the COVID-19 pandemic progresses, widespread community transmission of SARS-CoV-2 has ushered in a volatile era of viral immune evasion rather than the much-heralded stability of "endemicity" or "herd immunity." At this point, an array of viral strains has rendered essentially all monoclonal antibody therapeutics obsolete and strongly undermined the impact of vaccinal immunity on SARS-CoV-2 transmission. In this work, we demonstrate that antibody escape resulting in evasion of pre-existing immunity is highly evolutionarily favored and likely to cause waves of short-term transmission. In the long-term, invading strains that induce weak cross-immunity against pre-existing strains may co-circulate with those pre-existing strains. This would result in the formation of serotypes that increase disease burden, complicate SARS-CoV-2 control, and raise the potential for increases in viral virulence. Less durable immunity does not drive positive selection as a trait, but such strains may transmit at high levels if they establish. Overall, our results draw attention to the importance of inter-strain cross-immunity as a driver of transmission trends and the importance of early immune evasion data to predict the trajectory of the pandemic.

## Introduction

Some early commentators bullishly predicted the end of the COVID-19 pandemic [1–4], with the build-up of vaccine and natural immunity eventually curtailing SARS-CoV-2 transmission. However, the pandemic is now entering its fourth year despite a vast burden of prior infection, over 13 billion vaccination doses globally [5] and high prevalence of anti-SARS-CoV-2 antibodies [6,7]. Consistent with early-pandemic warnings [8–13], the pace of immune evasion has proven rapid [14,15], and transmission has continued robustly in the post-vaccine era [16]. Warnings about insufficient vaccine acceptance [17], rapid waning of vaccine and post-

Schematic Fig 1) are available in the following Github repository: https://github.com/madistod/two-strain-SEIR/tree/main.

**Funding:** L.F.W. received funding through National Institutes of Health (nih.gov), award number R35GM141821. The funder had no role in the study design, data analysis, decision to publish or preparation of the manuscript.

**Competing interests:** MS and AC are employees of, and shareholders of, Fractal Therapeutics. Fractal Therapeutics has no commercial interest in the subject of this paper.

infection immune protection [18,19], and antibody evasion [8–10,20] have all materialized at this point [21–23], leading to the high levels of SARS-CoV-2 transmission.

The ability of SARS-CoV-2 to evade immunity through mutations that degrade neutralizing antibody binding has been a major driver of ongoing viral transmission, as neutralizing antibodies have been demonstrated to be the correlate of immune protection for SARS-CoV-2 [24–26]. Indeed, the post-omicron era of the pandemic has been marked by successive waves driven by immune-evading strains including BA.1, BA.5, XBB, and XBB.1.5 [14,22,27–29]. These immune-evading strains acquire an evolutionary advantage in the context of widespread immunity through mutations that degrade the binding of neutralizing antibodies induced by infection with prior strains or by vaccines (antibody escape through "antigenic drift") [30]. As neutralizing antibodies mediate sterilizing immunity to SARS-CoV-2–that is, they block infection upon exposure—evasion of these antibodies promotes reinfection [31]. This advantage has allowed these immune-evading strains to achieve dominance, drive spikes in transmission, and replace (succeed) pre-existing strains [20]. Between December 2021 (initial dissemination of the original BA.1 omicron) and December 2022, several strain succession events were documented resulting in an approximately 35-fold loss in neutralizing titer [32].

Understanding the potential of emerging strains to drive waves of infection, persist in circulation, and co-circulate with pre-existing strains is vital for understanding and reacting to this volatile phase of the pandemic. Immune evasion has implications for short-term and long-term transmission levels [33], possible changes in disease severity [34], and the efficacy of vaccines and therapeutics, especially monoclonal antibodies [22,35]. Anticipating the behavior of viral strains has tremendous practical significance for designing nonpharmaceutical and biomedical interventions.

In this study, we use an epidemiological modeling framework to build a quantitative understanding of the role of immune evasion in inter-strain competition and selection dynamics under endemic conditions. To this end, we developed a two-strain Susceptible- Infectious-Recovered- Susceptible (SIRS) model accounting for variable cross-immunity between a pre-existing and an invading strain. This paper explores viral evolutionary strategies by simulating a few relevant immunological scenarios: antigenic drift, which we surmise may result in symmetric or unilateral antibody escape, and induction of less durable immunity.

Antigenic drift results in reduced cross-immunity (immunity induced by one strain against another) compared to homologous immunity (immunity induced by a strain against itself) [36]. If the impact of antigenic drift is symmetric, the invading strain's cross immunity against the original strain will equal the original strain's cross immunity against the invading strain. The plausibility of this scenario is supported by the tolerance of SARS-CoV-2's spike for a wide variety of mutations [8,37,38]. However, omicron BA.1 appeared to benefit from essentially unilateral antibody escape: while BA.1 strongly evaded pre-existing immunity to delta, delta was impeded by immunity induced by BA.1 [39]. The final scenario regards the possibility of viral strains with reduced durability of immunological protection from reinfection. Possibly exemplifying this scenario is omicron, which appears to exert weaker protection against homologous reinfection than delta (prior omicron reduces risk of omicron reinfection by 59.3%; prior delta infection reduces risk of delta reinfection by 92.3% [40].)

Determining the immunological properties likely to be selected for is crucial for predicting the trajectory of SARS-CoV-2 under widespread transmission. Although the rapid pace of SARS-CoV-2 evolution and the simultaneous emergence of immune-evading multiple strains paints a complex picture [41,42], this simplified analysis provides a basis for understanding the inter-strain competition and selection that underpin these dynamics. Identifying the characteristics of strains likely to be successful and drive significant waves of transmission is crucial to support early-warning systems. Co-circulation of viral serotypes–that is, viral strains

sufficiently antigenically distinct to coexist [43]–is an emergent threat that requires greater understanding and may lead to dramatically worse outcomes in the long-term trajectory of the pandemic.

## Methods

To evaluate short-term and long-term transmission of novel strains of SARS-CoV-2, we built a two-strain SIRS (susceptible-infectious-recovered-susceptible) model represented by a series of ordinary differential equations (ODEs). We implemented the model in Python using Scipy's solve_ivp ODE solver [44]. Relevant code has been uploaded to a Github repository.

The model has two sets of parallel infected and recovered compartments, representing those infected with and recovered from the original strain and the invading strain. While an individual may possess immunity to both strains simultaneously, we excluded the possibility of co-infection. The model considers disease transmission, waning immunity, and induction of immunity due to exposure to either strain or both strains. We assumed a constant population size, such that birth rate is equal to death rate. We did not account for disease-related deaths. Fig 1 is a visual representation of the model compartments and transitions between compartments through infection, recovery, and immune waning.

The model parameters that are fixed among all model iterations are listed in Table 1A; parameters varied during the analysis and their ranges are described in Table 1B. The US population size and birth rate are used to inform the model. However, birth rate is expected to have minimal impact on immunological dynamics for frequent reinfection pathogen such as SARS-CoV-2, and the SIRS model scales linearly with population size [45]. As a result, the per-capita infection rates predicted by our model are expected to extend well to other populations.

In Table 1B, variable parameter ranges were chosen to reflect the range of plausible values. For example, commonly cited estimates for the intrinsic reproductive number ($R_0$) of measles–which is highly contagious–reach up to 18 [52]. As a result, we explored a range of $R_0$ values from 0 to 20 to address a wide range of possible scenarios. We varied cross-immunity values from none (0) to complete (1). The waning factor represents a fold-increase in the waning rate of immunity, which we varied from 1-fold to 10-fold to represent a wide range of possibilities.

The effective infectiousness for the original and invading strains, $\beta_O$ and $\beta_I$, respectively, are derived from the relationship between $R_{0,O}$ and $R_{0,I}$ and the recovery rate, $\gamma$. $\beta_O$ and $\beta_I$ are calculated as follows:

$$\beta_O = \gamma \, R_{0,O}$$

$$\beta_I = \gamma \, R_{0,I}$$

The effective infectiousness for reinfection, $\beta_{RO}$, and $\beta_{RI}$ can be derived from the relationship between the effective infectiousness and cross-immunity parameter, C:

$$\beta_{RO} = \beta_O * (1 - C_{OvI})$$

$$\beta_{RI} = \beta_I * (1 - C_{IvO})$$

To represent symmetric antibody escape, the cross-immunity of the original strain against the invading strain ($C_{OvI}$) is set equal to the cross-immunity of the invading strain against the original strain ($C_{IvO}$). In the shorter duration of immunity scenario, there is assumed to be complete cross-immunity between the two strains; as a result, no individuals are simultaneously immune to both strains (no individuals in the $R_{OI}$ compartment).

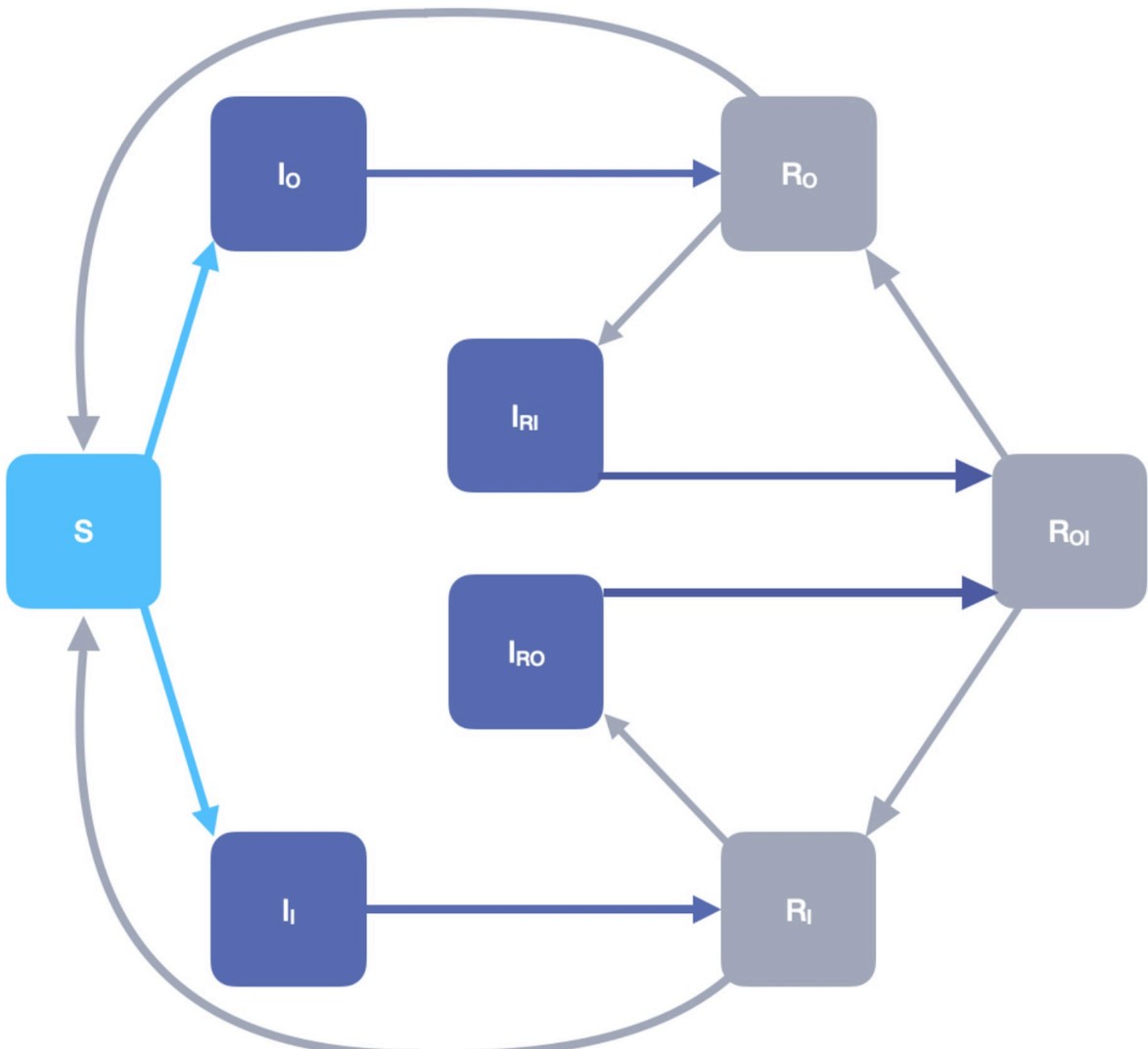

**Fig 1. Schematic representing the two-strain SIRS model.** Light blue represents fully susceptible individuals, dark blue represents infected individuals, and gray represents recovered individuals.

The ODEs for the model are summarized below:

$$\frac{dS}{dt} = \mu N - \frac{\beta_O S I_O}{N} - \frac{\beta_O S I_{rO}}{N} - \frac{\beta_I S I_I}{N} - \frac{\beta_I S I_{rI}}{N} + \rho R_O + \tau \rho R_I - \mu S$$

$$\frac{dI_O}{dt} = \frac{\beta_O S I_O}{N} + \frac{\beta_O S I_{rO}}{N} - \gamma I_O - \mu I_O$$

$$\frac{dI_I}{dt} = \frac{\beta_I S I_I}{N} - \frac{\beta_I S I_{rI}}{N} - \gamma I_I - \mu I_I$$

**Table 1. A: Model parameters fixed between scenarios.** B: Variable parameters and their values by scenario.

| Symbol | Parameter | Value | Units |
|---|---|---|---|
| $\mu$ | Birth rate | $2.5 \times 10^{-5}$ [46] | 1/person/days |
| $N$ | Population size | 330,000,000 [47] | Individuals |
| $\rho$ | Immune waning rate | 0.0018 [48] | 1/days |
| $\gamma$ | Recovery rate | 0.1 [49] | 1/days |
| $R_{0,O}$ | Intrinsic reproductive number, original strain | 8.2 [50,51] | Individuals |

| Symbol | Parameter | Value, symmetric antibody escape | Value, unilateral antibody escape | Value, shorter duration of immunity | Units |
|---|---|---|---|---|---|
| $R_{0,I}$ | Intrinsic reproductive number, invading strain | 0–20 | 0–20 | 0–20 | individuals |
| $C_{OvI}$ | Cross-immunity of original strain against invading strain | 0–1 | 0–1 | 1 | n/a |
| $C_{IvO}$ | Cross-immunity of invading strain against original strain | 1 | 0–1 | 1 | n/a |
| $\tau$ | Waning factor | 1 | 1 | 1–10 | n/a |

The immune waning rate of 0.0018/day corresponds to a mean duration of protection from infection of 550 days.

$$\frac{dI_{rO}}{dt} = \frac{\beta_{rO}R_I I_O}{N} + \frac{\beta_{rO}R_I I_{rO}}{N} - \gamma I_{rO} - \mu I_{rO}$$

$$\frac{dI_{rI}}{dt} = \frac{\beta_{rI}R_O I_I}{N} + \frac{\beta_{rI}R_O I_{rI}}{N} - \gamma I_{rI} - \mu I_{rI}$$

$$\frac{dR_O}{dt} = \gamma I_O - \frac{\beta_{rI}R_O I_I}{N} - \frac{\beta_{rI}R_O I_{rI}}{N} - \rho R_O + \tau\rho R_{OI} - \mu R_O$$

$$\frac{dR_I}{dt} = \gamma I_I - \frac{\beta_{rO}R_I I_O}{N} - \frac{\beta_{rO}R_I I_{rO}}{N} - \tau\rho R_I + \rho R_{OI} - \mu R_I$$

$$\frac{dR_{OI}}{dt} = \gamma I_{rO} + \gamma I_{rI} - \rho R_{OI} - \tau\rho R_{OI} - \mu R_{OI}$$

Where S is the susceptible population, I is the infectious population, and R is the recovered population. Subscripts O and I represent the original strain and the invading strain, respectively. Individuals in the $R_{OI}$ compartment have immunity against both strains. $I_{rO}$ represents individuals who are infected with the original strain while immune to the invading strain; $I_{rI}$ represents individuals who are infected with the invading strain while immune to the original strain.

Initial conditions for the model are set such that the original strain is at an endemic steady-state, meaning the number of active infections is constant. The steady-state distribution of individuals in each compartment was determined by running the simulation for 10,000 days in the absence of the invading strain. Using this method, we found that the steady-state number of active original strain infections was $5.27 \times 10^6$, with $2.84 \times 10^8$ individuals recovered. To simulate invasion, we set the initial value of $I_I$—infections with the invading strain—to one.

We used the model to assess the invading strain's ability to invade and drive a wave of transmission in the short-term as well as the impact of invading strains on long-term transmission trends. We define an outbreak as successful if the invading strain does not go extinct. We evaluated short-term transmission over a six-month period to reflect the period of enhanced transmission before immunity induced by the novel strain constrains its spread. To determine the long-term impact of strain invasion on steady-state SARS-CoV-2 transmission levels, we ran the two-strain model for 10 years, after which point immunological equilibrium is reached. Yearly infections are read out over the last year of the simulation.

## Results

### Mutual immune evasion favors successful invasion and co-circulation

In Figs 2 and 3, we explore epidemiological outcomes after the introduction of an invader strain with mutually reduced cross-immunity with respect to the original strain. We assumed immune evasion is perfectly symmetric: that is, the invader strain's sensitivity to immunity induced by the original strain is equal to the original strain's sensitivity to new immunity induced by the invader strain. Fig 2 demonstrates that if immune evasion is symmetric, invasion is successful under a wide range of transmissibility and immune evasion conditions. Even a poorly transmissible strain relative to currently circulating strains may successfully invade if cross-immunity is low enough. This suggests that symmetric immune evasion is highly evolutionarily favorable as a trait, and selection for symmetric immune evasion may overcome deficits in intrinsic reproductive number.

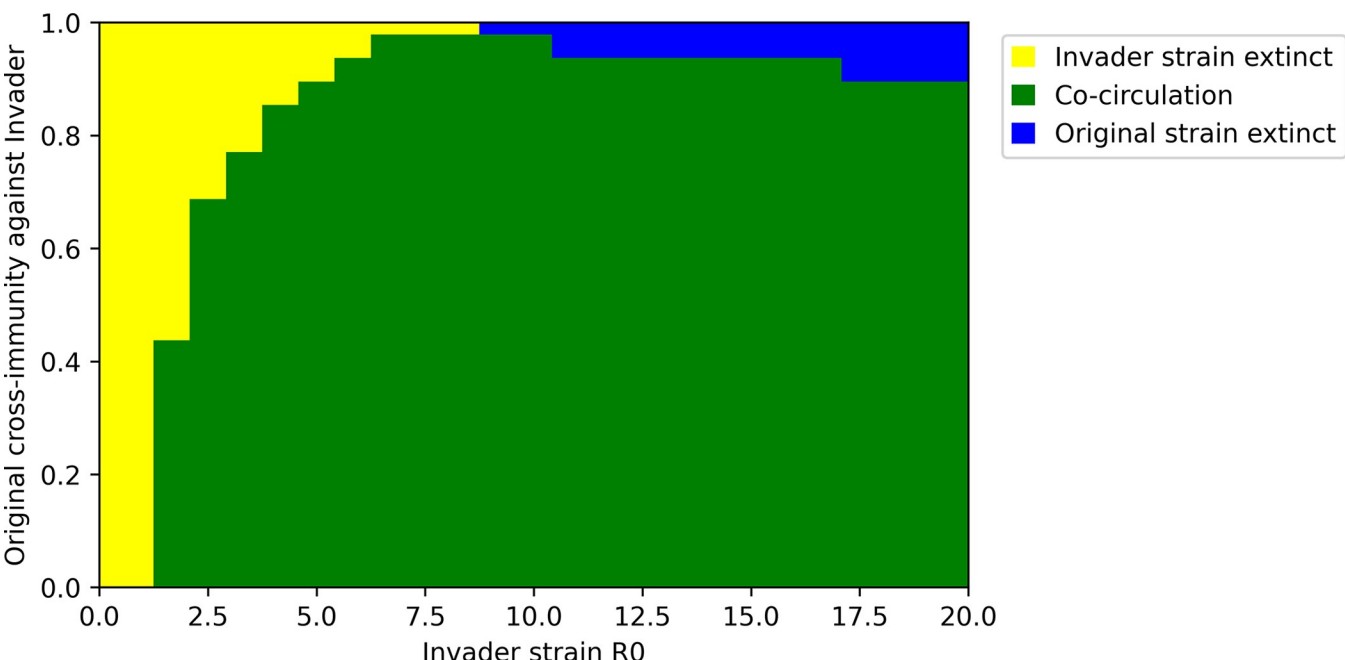

**Fig 2. Invasion outcomes for invader strains with varying transmissibility ($R_0$) and symmetric cross-immunity with the original strain.** Outcomes are classified based on whether either strain becomes extinct or the strains co-circulate. Invasion is successful unless the invader strain becomes extinct (yellow).

Additionally, if immune evasion is strong and symmetric, the success of the invader strain may not come at the expense of the original strain. Succession (extinction of the original strain) is expected only in scenarios where cross-immunity between the strains is high, and the invading strain is more transmissible than the original strain.

### Invasion increases transmission in the short-term and co-circulation increases transmission in the long-term

Fig 3 explores short-term and long-term strain transmission of the original and invader strains after various invasion scenarios. As shown in Fig 3A, invasion of a novel strain may only modestly reduce transmission of the original strain in the short-term, while extensive transmission of the invading strain may occur (Fig 3B). This results in a large increase in overall transmission in the short-term (a "wave" or a "spike") for successful invasion scenarios compared to failed invasions, which reflect baseline transmission of the original strain (Fig 3C). Interestingly, short-term reductions in transmission of the original strain upon invasion (Fig 3A) do not always result in extinction of the original strain (Fig 2). In general, higher transmissibility of the invader strain and greater mutual immune evasion (lower cross-immunity) promote larger waves of transmission.

In the long-term, when steady-state is reached, low symmetric cross-immunity that results in co-circulation of strains causes the highest overall transmission (Fig 3D–3F.) This is because as cross-immunity levels decrease, competition between the strains also decreases. We also note that increases in transmissibility in the absence of immune evasion–represented by the uppermost section of each plot where cross-immunity approaches one–are unlikely to drive significant increases in transmission in the short- or long-term. However, overall transmission remains high in these scenarios, nearing 60% of the population infected annually (Fig 3F).

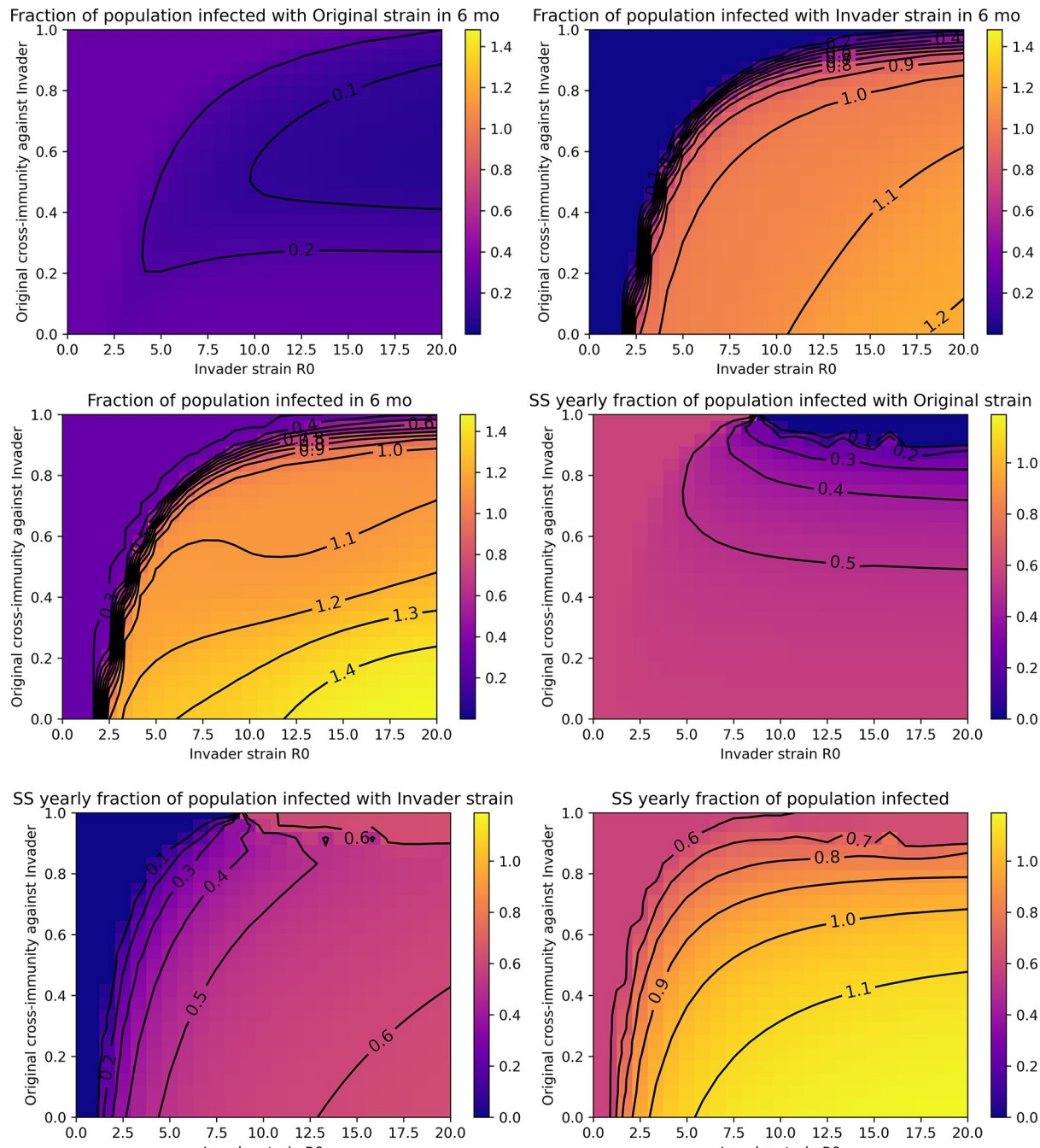

**Fig 3. Short-term and long-term transmission trends after introduction of invader strains with varying transmissibility ($R_0$) and symmetric cross-immunity properties.** The total number of infections is expressed as a fraction of the population size (represented by the colorbar). Fractions greater than 1 indicate reinfections. Total infections over the first six months after invader strain introduction due to **A)** the original strain, **B)** the invader strain, or **C)** both summed together. Total infections over one year at steady-state (SS) for **D)** the original strain, **E)** the invader strain, or **F)** both summed together. Colormaps are scale-matched across subpanels A-C and D-F.

## Unilateral immune evasion favors succession

Fig 4 explores invasion scenarios in which cross-immunity is unilateral: the invader strain evades the original strain's immunity but exerts full cross-immunity against the original strain.

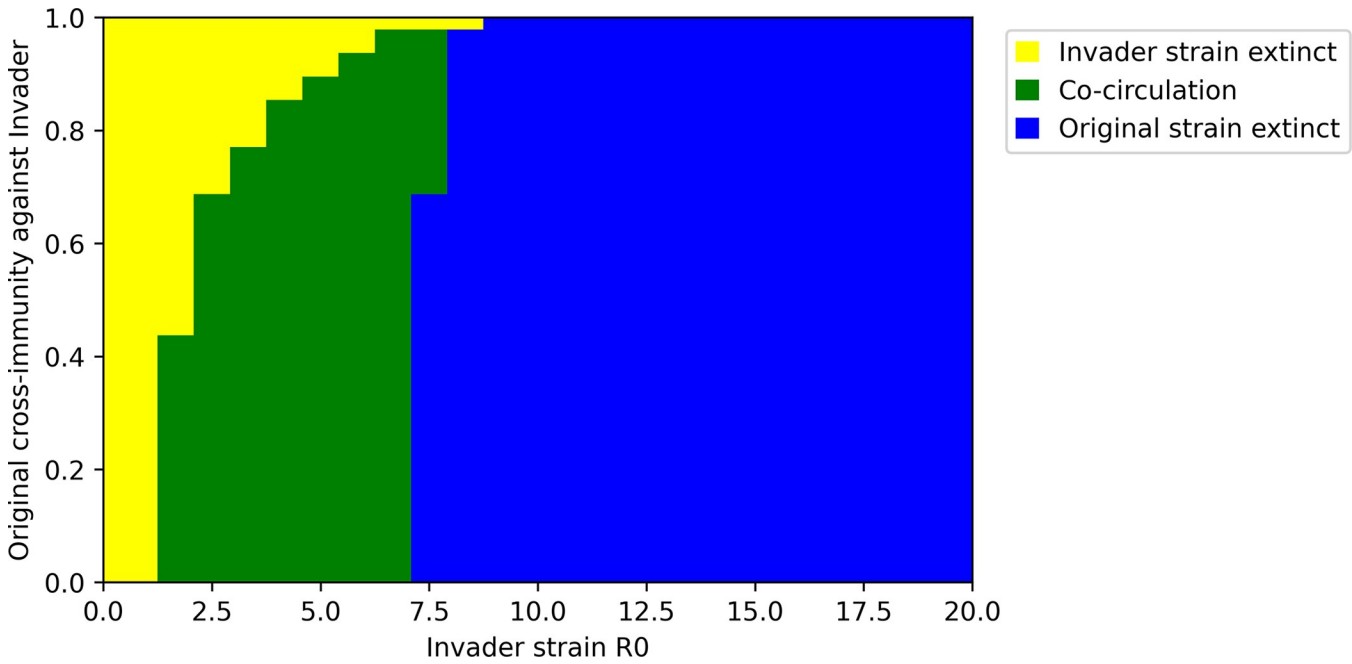

**Fig 4. Invasion outcomes for novel strains with varying transmissibility ($R_0$) and degrees of unilateral immune evasion.**

While Fig 2 demonstrates that succession (extinction of the original strain) is rare if immune evasion is perfectly symmetric, Fig 4 indicates that unilateral immune evasion favors succession, especially if the $R_0$ of the invader strain is equal to or greater than the original strain's $R_0$. However, the conditions under which invasion succeeds–represented by both co-circulation and original strain extinction outcomes–are identical to the symmetric immune evasion scenario.

In the short-term, transmission dynamics after emergence of an invading strain in the unilateral immune evasion scenario are similar to the symmetric scenario (Fig 5A–5C). This is intuitive: the difference between the two scenarios is in the nature of the invading strain's immunity, which is not yet present in the short-term. In the long-term, immunity created by the invading strain constrains spread of the original strain, in many cases leading to its extinction (Fig 5D–5F). This reduces transmission of the original strain with limited benefit to the invading strain, except in cases where the invading strain's dominance is weaker. As a result, long-term, overall transmission is not substantially impacted by successful invasion when cross-immunity induced by the invading strain against the original strain is strong.

### Duration of invader strain's immunity does not impact its potential to invade

We also simulated invasion attempts by strains that exert shorter-term immunity. We found that in the absence of neutralizing antibody evasion, the duration of immunity induced by the invading strain does not impact its ability to establish (Fig 6). This is consistent with the intuition that immunity induced by the invader strain does not yet exist at the time of invasion and likewise does not impact invasion success. Additionally, a strain that induces less durable immunity achieves no selective advantage in the absence of immune evasion because the shorter immunity increases the reinfection potential of both strains equally. Additionally, we note that variation in this property in the absence of immune evasion does not support

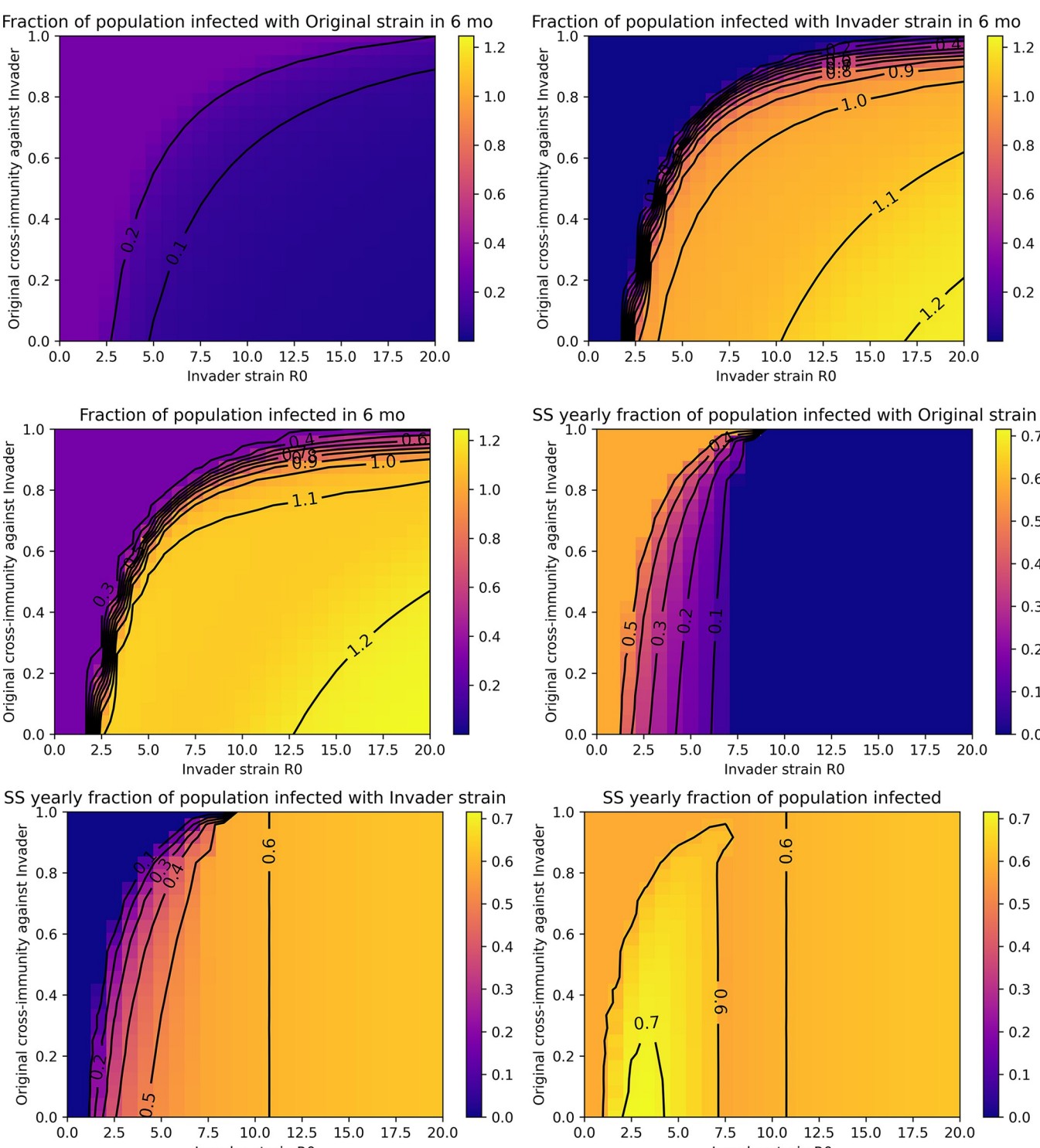

**Fig 5. Short-term and long-term transmission trends after introduction of invader strains with varying transmissibility ($R_0$) and unilateral cross-immunity properties.** In these scenarios, the invader strain induces strong immunity against the original strain, while the original strain induces weaker immunity against the invader strain. The total number of infections is expressed as a fraction of the population size. Fractions greater than 1 indicate reinfections. Total infections over six months due to **A)** the original strain, **B)** the invader strain, or **C)** both summed together. Total infections over one year at steady-state for **D)** the original strain, **E)** the invader strain, or **F)** both summed together. Colormaps are scale-matched across subpanels A-C and D-F.

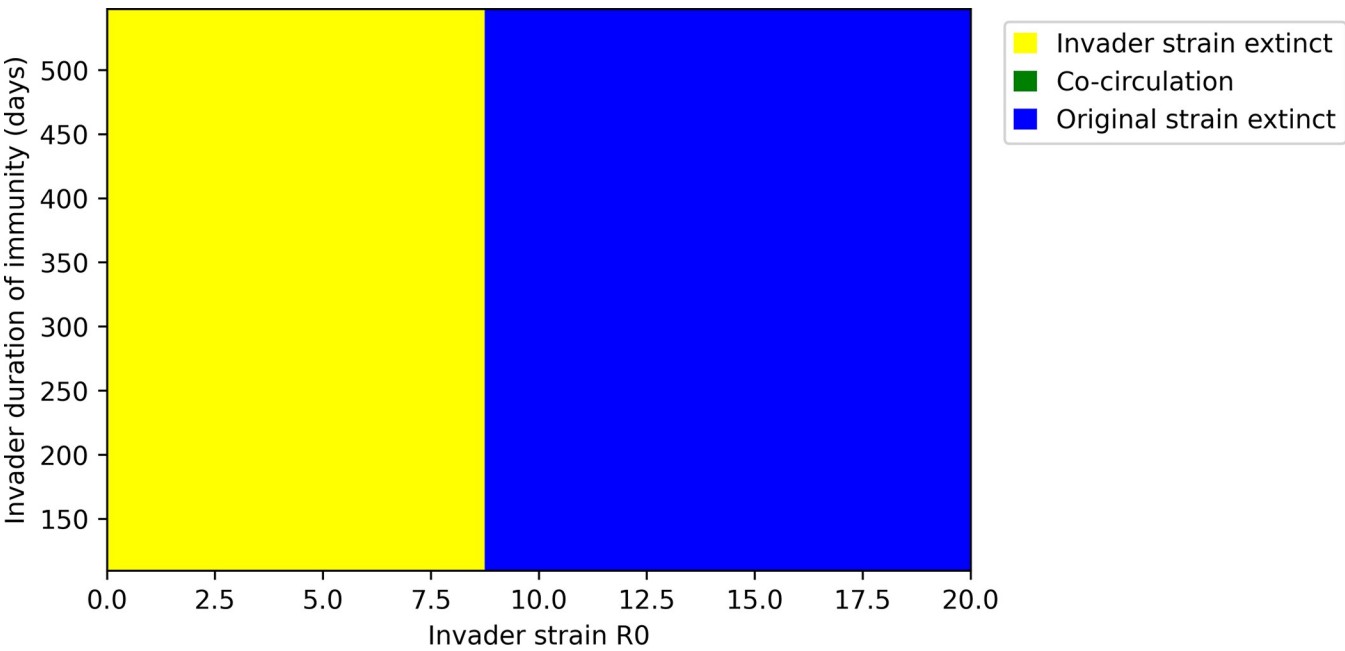

**Fig 6. Outcomes for invasion scenarios in which the invader strain induces less durable immunity than the original strain.** The duration of immunity against wild-type SARS-CoV-2 infection is estimated to be 550 days [48].

coexistence of strains. However, strains with shorter duration of immunity may transmit at much higher levels in the long term (S1 Fig).

## Discussion

The epidemiological modeling analyses in this work demonstrate that cross-immunity between circulating and emergent strains is expected to be a major driver of the ability of invading strains to successfully establish as well as short-term and long-term transmission trends. This includes the possibility of stable co-circulation of antigenically distinct strains (serotypes) [Figs 2 & 4]. Under conditions of widespread anti-SARS-CoV-2 immunity, neutralizing antibody evasion likely provides an evolutionary advantage. Increases in immune evasiveness can be expected to drive selection and transmission to a greater extent than further increases in intrinsic transmissibility ($R_0$) alone [Figs 3 & 5].

In this work, we showed that the intrinsic transmissibility of the emerging strain and the immunological landscape a new strain encounters (by definition, imposed by pre-existing strains) shape its invasion potential and extent of transmission in the short-term. This is the cross-immunity of the original strain against the invader strain. Strains with antibody escape that weakens pre-existing immunity are likely to be strongly selected for and drive significant increases in short-term transmission [Figs 3A–3C and 5A–5C]. In this way, ongoing viral evolution is likely to contribute to long-term high and volatile levels of SARS-CoV-2 transmission. Notably, our work shows that even poorly transmissible (low $R_0$) strains may successfully invade if they sufficiently evade pre-existing neutralizing antibodies.

Our analyses further showed that while evasion of pre-existing immunity dictates which strains will successfully emerge, the properties of immunity induced by successful invading strains contribute to their long-term impact on overall transmission levels. For example, we have shown that invader strains inducing less durable immunity can sustain greater long-term transmission if invasion is successful [S1 Fig].

Additionally, we have shown here that weak cross-immunity of the invader strain against the original strain can permit the invader and original strains to coexist with each other and continue to co-circulate [Figs 2 & 4]. This situation, referred to as serotype formation [53], may significantly increase long-term transmission levels. The existence of serotypes for other human pathogens (e.g. dengue) has made their control more complicated, as tests and vaccines have to be created to match each serotype [54]. Serotypes have already been observed for human coronaviruses OC43 and NL63 [55,56]. Co-circulation of SARS-CoV-2 serotypes would increase demand on already-strained global sequencing systems for tracking viral evolution and may exacerbate the challenge of matching vaccines and therapeutics to circulating strains. Serotypes are also thought to increase the risk of antibody-dependent enhancement (ADE), as has been observed for dengue [57]. Although ADE is not thought to be a risk for SARS-CoV-2 at present, the co-circulation of two or more SARS-CoV-2 serotypes would create conditions known to contribute to ADE in other viral diseases [58].

These findings highlight the importance of measuring cross-immunities between pre-existing and emergent strains to forecast the likelihood of invasion success and short-term and long-term implications of a successful invasion. Although real-world cross-immunity is challenging to measure, neutralizing potency of post-immunization sera has proven to be a practical correlate [59]. Many studies assess the neutralization potency induced by vaccines against emerging strains [60,61], while some studies have also assessed the neutralization potency raised by circulating strains against emerging strains [62–64]. Using models linking neutralizing antibody potency to extent of protection, this data can be leveraged to estimate cross-immunity against invading strains [25]. This is important for predicting which strains are likely to establish successfully and significantly increase short-term transmission.

However, data is often lacking regarding the neutralization potency induced by emergent strains against pre-existing strains. Measuring the neutralizing potency of invading strain infectee sera against pre-existing strains could reveal the risk of long-term co-circulation. In this work, we have demonstrated that co-circulation may follow apparent declines in the transmission of the original strain during high post-emergence transmission of the invader strain. This underlines the importance of *in vitro* or clinical data regarding cross-immunity of invading strains against pre-existing strains. The likelihood of co-circulation may not be obvious in early epidemiological data, at which point interventions are most likely to succeed [65].

There are a few limitations to this analysis. Firstly, we did not simulate the impact of vaccines in this analysis. In our prior work [66], we have explored the impact of vaccines on SARS-CoV-2 evolutionary dynamics; we determined that the impact of vaccines on evolutionary dynamics is limited due to minimal vaccinal protection against infection. Secondly, our SIRS model implementation categorizes individuals into four immunological groups: fully susceptible, immune to original strain and susceptible to invader strain, immune to invader strain and susceptible to original strain, and immune to both strains. As we have explored in other analyses using agent-based models, the immunity landscape is more complex than this SIRS model can capture. This complexity arises due to heterogeneity in individual exposure to infection and vaccination, interindividual variability in neutralizing antibody durability, and neutralizing antibody build-up over successive infections and vaccinations [48,67]. In particular, we surmise that the build-up of neutralizing potency over successive infections may cause the apparent cross-immunities between strains to not be fixed over time. This may result in additional competitive dynamics in the long term. Lastly, our analysis addresses competition between two strains for hosts in short and long timeframes. In reality, multiple strains may be in competition simultaneously, and the emergence of further invader strains may occur before the "long-term" two-strain dynamics explored here are fully realized [68,69]. We also acknowledge the possibility of short-term "pseudo" co-circulation, in which strains may appear to co-

circulate in the short-term before sufficient infections have occurred to impose immunological constraints. Nevertheless, our work provides a basis for understanding basic competition dynamics between any two strains.

With these limitations in mind, the purpose of this analysis is to broadly identify strain properties conducive to invasion and to highlight the risk and implications of co-circulation of SARS-CoV-2 strains. Our work demonstrates that the emergence of immune-evading strains is a key driver of transmission rates at this stage of the pandemic, as invasion events are likely to drive extensive transmission in the short-term. We emphasize the value of early data characterizing the antibody evasion of and immune responses to emergent strains to inform SARS-CoV-2 vaccination and other mitigation strategies.

Our work also suggests the emergence of SARS-CoV-2 serotypes as a further potential risk of the current public health strategy. Stable co-existence of strains has not yet been demonstrated at the time of writing, perhaps due to strong cross-immunity against original strains as demonstrated with Delta and Omicron BA.1 [39]. In sum, the results presented here point to the likelihood of high baseline SARS-CoV-2 transmission, with the possibility of transient increases in transmission ("waves") driven by successive (or co-circulating) waves of immune-evasive strains. These predictions are reflected in the real-world emergence and widespread circulation of many competing immune-evasive SARS-CoV-2 strains [41,42]. We identify inter-strain cross-immunity as an important variable for predicting outcomes and gauging risk in this seemingly unpredictable scenario.

## Supporting information

**S1 Fig. Short-term and long-term transmission trends after introduction of invader strains with varying transmissibility ($R_0$) and durability of immunity.** The total number of infections is expressed as a fraction of the population size. Fractions greater than 1 indicate reinfection. Total infections over six months due to **A)** the original strain, **B)** the invader strain, or **C)** both summed together. Total infections over one year at steady-state for **D)** the original strain, **E)** the invader strain, or **F)** both summed together. Colormaps are matched across subpanels A-C and D-F.
(ZIP)

## Author Contributions

**Conceptualization:** Debra Van Egeren, Arijit Chakravarty, Laura F. White, Madison Stoddard.

**Data curation:** Catherine Albright, Madison Stoddard.

**Formal analysis:** Catherine Albright, Madison Stoddard.

**Funding acquisition:** Laura F. White.

**Investigation:** Catherine Albright, Madison Stoddard.

**Methodology:** Catherine Albright, Aditya Thakur, Madison Stoddard.

**Project administration:** Madison Stoddard.

**Software:** Madison Stoddard.

**Supervision:** Arijit Chakravarty, Laura F. White, Madison Stoddard.

**Validation:** Madison Stoddard.

**Visualization:** Catherine Albright, Madison Stoddard.

**Writing – original draft:** Catherine Albright, Madison Stoddard.

**Writing – review & editing:** Catherine Albright, Debra Van Egeren, Aditya Thakur, Arijit Chakravarty, Laura F. White, Madison Stoddard.

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
