## [Decision Letter · Decision Letter 0]

18 Jul 2023

PONE-D-23-18143Antibody escape, the risk of serotype formation, and rapid immune waning: modeling the implications of SARS-CoV-2 immune evasionPLOS ONE

Dear Dr. Stoddard,

Thank you for submitting your manuscript to PLOS ONE. After careful consideration, we feel that it has merit but does not fully meet PLOS ONE’s publication criteria as it currently stands. Therefore, we invite you to submit a revised version of the manuscript that addresses the points raised during the review process.

Kindly look through the Reviewers' comments and attend to the major subjects raised. Ensure to provide detailed responses and corrections are required.

We look forward to receiving your revised manuscript.

Kind regards,

Olatunji Matthew Kolawole, Ph.D.

Academic Editor

PLOS ONE

Journal Requirements:

'MS and AC are employees of, and shareholders of, Fractal Therapeutics. Fractal Therapeutics has no commercial interest in the subject of this paper."

Reviewers' comments:

Reviewer's Responses to Questions

**Comments to the Author**

1. Is the manuscript technically sound, and do the data support the conclusions?

Reviewer #1: Yes

Reviewer #2: Yes

Reviewer #3: Yes

Reviewer #4: Partly

Reviewer #5: Yes

2. Has the statistical analysis been performed appropriately and rigorously? 

Reviewer #1: Yes

Reviewer #2: Yes

Reviewer #3: N/A

Reviewer #4: N/A

Reviewer #5: Yes

3. Have the authors made all data underlying the findings in their manuscript fully available?

Reviewer #1: Yes

Reviewer #2: Yes

Reviewer #3: Yes

Reviewer #4: Yes

Reviewer #5: Yes

4. Is the manuscript presented in an intelligible fashion and written in standard English?

Reviewer #1: Yes

Reviewer #2: Yes

Reviewer #3: Yes

Reviewer #4: Yes

Reviewer #5: Yes

5. Review Comments to the Author

Reviewer #1: Beyond looking at the infection and invasion models in block, it will be good take it to the level of the targeted antibodies (Spike, RBD and N). This will make your points more focused particularly when discussing the waning of SARS-COV-2 antibodies.

References:

1. Kellam P, Barclay W. The dynamics of humoral immune responses following SARS-CoV-2 infection and the potential for reinfection. J Gen Virol. 2020 Aug;101(8):791-797. doi: 10.1099/jgv.0.001439. PMID: 32430094; PMCID: PMC7641391

2. Pérez-Alós, L., Armenteros, J. J. A., Madsen, J. R., Hansen, C. B., Jarlhelt, I., Hamm, S. R., ... & Garred, P. (2022). Modeling of waning immunity after SARS-CoV-2 vaccination and influencing factors. Nature communications, 13(1), 1614.

Reviewer #2: This article was intelligently written. Though I am not a statistican and I am not into epidemological modelling, but the article was something I could relate with a bit. The research was original and could be published.

Reviewer #3: You emphasize the value of early data characterizing the antibody evasion of and immune responses to emergent variants to inform SARS-CoV-2 vaccination and other mitigation strategies. Your work also suggests the emergence of SARS-CoV-2 serotypes as a further potential risk of the current public health strategy.

Reviewer #4: The study is relevant, original, and based on a current topic of global importance. The rationale is clear and valid. However, some changes are necessary:

1. Table 1A uses data on the population of the United States in 2023, with the birth rate for 2018 (pre-pandemic period), with data on COVID-19 for the year 2022. I encourage the authors to discuss how much this simulation can be extrapolated to other countries and populations. In Table 1B, however, the values defined for each variable were not referenced or explained.

2. The methodology does not inform, for example, the software(s) used to carry out the analyses. In general, the execution of the procedures should be described in greater detail.

3. There was a lack of a clear explanation of the motivation behind the analyzes carried out and the parameters that were chosen/defined. (For example, a period of 180 days to quantitatively investigate the success of an outbreak should be referenced or at least explained).

4. The exposition of results and discussion has many assertions based on hypotheses or authorial inference without sufficient data to support them.

5. The three first paragraphs of the discussion do not have any references. In the next paragraph, the text highlighted the possibility of SARS-CoV-2 causing ADE, which is not the focus of this paper, nor are there robust results that support this kind of debate.e

6. Figures need better resolution and be more self-explanatory. In addition, all acronyms used in the images must appear in the caption.

7. The world has already witnessed changes in the strains of importance. The discussion section could approach how much the simulations carried out coincide with the patterns already observed.

After these adjustments, I believe the article will be suitable for publication.

Reviewer #5: It is an innovative study and offers an interesting approach to pandemic preparedness. However, the development of the topic falls short of the title of the paper. The concept of antigenic drift and its relationship to the concepts of variant, strain and serotype should be revised and the wording adapted to this terminology. Reference should be made to the possible risk of serotyping and the idea of "dynamic competence" should be better supported.

6. PLOS authors have the option to publish the peer review history of their article (what does this mean?). If published, this will include your full peer review and any attached files.

Reviewer #1: **Yes: **Ado Garba Abubakar

Reviewer #2: **Yes: **Emmanuel Adamolekun

Reviewer #3: **Yes: **Dr Arifa Akram

Reviewer #4: No

Reviewer #5: **Yes: **Magda Yoana Beltrán León

---

## [Author Response · Author response to Decision Letter 0]

7 Sep 2023

In response to the editor's request for data availability, we have attached the Python code used to perform the analysis in a Github repository. The URL is provided with the manuscript files. All figures were generated using this code, with the exception of model schematic Fig 1. 

We have provided our response to reviewers in the attached file. The contents of our response to reviewers are copied below: 

Reviewer #1: 

"Beyond looking at the infection and invasion models in block, it will be good take it to the level of the targeted antibodies (Spike, RBD and N). This will make your points more focused particularly when discussing the waning of SARS-COV-2 antibodies."

We appreciate this input from the reviewer and agree that greater specification regarding the type of antibody is necessary. We added language to indicate that the focus of the paper is on neutralizing antibodies, and we added the following explanation for this focus [lines 40-41]:

“as neutralizing antibodies have been demonstrated to be the correlate of immune protection for SARS-CoV-2.”

and added the following references to the text to support this assertion:

https://www.ncbi.nlm.nih.gov/pmc/articles/PMC8706198/

https://www.thelancet.com/journals/lanmic/article/PIIS2666-5247(21)00267-6/fulltext 

https://www.nature.com/articles/s41591-021-01377-8

Incidentally, the RBD of the viral Spike protein dominates the neutralizing antibody response.

Reference:

https://www.sciencedirect.com/science/article/pii/S0092867420312344

Reviewer #2: 

"This article was intelligently written. Though I am not a statistican and I am not into epidemiological modelling, but the article was something I could relate with a bit. The research was original and could be published."

We thank the reviewer for their kind words. We hope the article will be of interest to readers of many backgrounds.

Reviewer #3: 

"You emphasize the value of early data characterizing the antibody evasion of and immune responses to emergent variants to inform SARS-CoV-2 vaccination and other mitigation strategies. Your work also suggests the emergence of SARS-CoV-2 serotypes as a further potential risk of the current public health strategy."

We agree with the reviewer’s summary of the key points of our manuscript. We hope our work will inform public health management of SARS-CoV-2 and scientific surveillance. 

Reviewer #4: 

"The study is relevant, original, and based on a current topic of global importance. The rationale is clear and valid. However, some changes are necessary:"

1. "Table 1A uses data on the population of the United States in 2023, with the birth rate for 2018 (pre-pandemic period), with data on COVID-19 for the year 2022. I encourage the authors to discuss how much this simulation can be extrapolated to other countries and populations. In Table 1B, however, the values defined for each variable were not referenced or explained."

We thank the reviewer for this helpful suggestion. We added clarification regarding the behavior of SIRS models for populations of different sizes and birth rates [lines 114-118]. We expect the model to extrapolate well to other populations and believe this clarification strengthens the paper.

The ranges for variable parameters in Table 1B were selected to represent a wide range of plausible scenarios; we added language to the methods section [lines 126-131] to explain this. 

2. "The methodology does not inform, for example, the software(s) used to carry out the analyses. In general, the execution of the procedures should be described in greater detail."

We agree with the reviewer’s assessment that the manuscript would benefit from more description of the modeling implementation. We added more information to the methods sections describing the software used, code availability, and the model structure [lines 98-103].

3. "There was a lack of a clear explanation of the motivation behind the analyzes carried out and the parameters that were chosen/defined. (For example, a period of 180 days to quantitatively investigate the success of an outbreak should be referenced or at least explained)."

We thank the reviewer for their feedback on this point. The underlying motivation for the analyses is laid out by us in the introduction [lines 57-62] in terms of practical significance. We have added additional text [lines 170-174], in order to more clearly tie the specific analyses performed by us to this big-picture objective. We have now clarified that the “short-term” simulations are intended to reflect the “first wave” of transmission of a novel strain (when the system is out of equilibrium) while “long-term” simulations reflect the model’s behavior after equilibrium between infected and susceptible populations is reached. We hope that this makes the choice clearer.

4. "The exposition of results and discussion has many assertions based on hypotheses or authorial inference without sufficient data to support them."

We thank the reviewer for this observation. Many of the assertions made in the Discussion are a restatement of the findings in the Results, with an expansion of the implications in each case. To clarify this, we have made numerous changes in the Discussion section (these are enumerated in detail in the response to the next comment). Any conclusions reached in the discussion have now been articulated in the results in the appropriate section with figure references. Additionally, we added language to clarify which conclusions were drawn based on the present analysis and added citations for any conclusions drawn based on the literature. We believe that this change tightens up the Discussion significantly.

5. "The three first paragraphs of the discussion do not have any references. In the next paragraph, the text highlighted the possibility of SARS-CoV-2 causing ADE, which is not the focus of this paper, nor are there robust results that support this kind of debate."

The first three paragraphs of the discussion are a summary of the main findings of this paper. To clarify this, we have added a brief introduction to each result discussed [lines 290, 307, 310, 312, 316]. We thank the reviewer for their observation and believe this change makes the discussion clearer.

Regarding ADE, we appreciate the reviewer’s concern. We clarified this section of the discussion to connect our findings (continued co-circulation of both strains) to serotype formation [line 318]. As serotype formation is known to increase the risk of ADE for other viruses, we have made this point clearer in the discussion to connect our findings to ADE risk [lines 326-327].

6. Figures need better resolution and be more self-explanatory. In addition, all acronyms used in the images must appear in the caption. 

"We have reuploaded the Figure files to improve resolution. Additionally, we have included more context in the figure captions and defined any acronyms (particularly R0 in reference to viral transmissibility). We thank the reviewer for these observations and believe this improves readability of the manuscript. "

7. "The world has already witnessed changes in the strains of importance. The discussion section could approach how much the simulations carried out coincide with the patterns already observed."

We thank the reviewer for this suggestion. To tie the work into real-world observations, we added content to the last paragraph of the discussion to address the current circulation of various immune evading viral strains and the observation that current strains appear to induce strong cross-immunity against pre-existing strains [lines 387-392]. 

"After these adjustments, I believe the article will be suitable for publication."

Reviewer #5: 

"It is an innovative study and offers an interesting approach to pandemic preparedness. However, the development of the topic falls short of the title of the paper. The concept of antigenic drift and its relationship to the concepts of variant, strain and serotype should be revised and the wording adapted to this terminology. Reference should be made to the possible risk of serotyping and the idea of "dynamic competence" should be better supported."

We thank the reviewer for these suggestions. 

To harmonize the terminology used throughout the paper we have now defined “antigenic drift” as the process by which viral antigens mutate and “antibody escape” as the immune evasion that results from this process. 

We also changed the manuscript to reference only “strains” throughout, as the manuscript is concerned with the evolutionary selection of immune-evading strains (rather than inconsequential variation). 

The relationship between serotypes and immune evasion is currently discussed at the end of the introduction, and we have added text in the Discussion to reiterate the definition of serotypes [316-318] and connect it explicitly to the findings in this paper. 

We are uncertain of the meaning of “dynamic competence” and have not discussed this concept at this time.

---

## [Editor Report · Decision Letter 1]

12 Sep 2023

Antibody escape, the risk of serotype formation, and rapid immune waning: modeling the implications of SARS-CoV-2 immune evasion

PONE-D-23-18143R1

Dear Dr. Stoddard,

We’re pleased to inform you that your manuscript has been judged scientifically suitable for publication and will be formally accepted for publication once it meets all outstanding technical requirements.

Kind regards,

Olatunji Matthew Kolawole, Ph.D.

Academic Editor

PLOS ONE
---

## [Editor Report · Acceptance letter]

25 Sep 2023

PONE-D-23-18143R1 

Antibody escape, the risk of serotype formation, and rapid immune waning: modeling the implications of SARS-CoV-2 immune evasion 

Dear Dr. Stoddard:

I'm pleased to inform you that your manuscript has been deemed suitable for publication in PLOS ONE. Congratulations! Your manuscript is now with our production department. 

Kind regards, 

on behalf of

Dr. Olatunji Matthew Kolawole 

Academic Editor

PLOS ONE